# An exploratory study of functional brain activation underlying response inhibition in major depressive disorder and borderline personality disorder

Cody Cane[1]*, Dean Carcone[1], Katherine Gardhouse[1], Andy C. H. Lee[1,2], Anthony C. Ruocco[1,2]

1 Department of Psychological Clinical Science, University of Toronto, Toronto, Ontario, Canada,
2 Department of Psychology, University of Toronto Scarborough, Scarborough, Ontario, Canada

☯ These authors contributed equally to this work.
* cody.cane@mail.utoronto.ca

**Data Availability Statement:** The data is available from OSF at https://osf.io/u6yxf/.

## Abstract

Cognitive control is associated with impulsive and harmful behaviours, such as substance abuse and suicidal behaviours, as well as major depressive disorder (MDD) and borderline personality disorder (BPD). The association between MDD and BPD is partially explained by shared pathological personality traits, which may be underpinned by aspects of cognitive control, such as response inhibition. The neural basis of response inhibition in MDD and BPD is not fully understood and could illuminate factors that differentiate between the disorders and that underlie individual differences in cross-cutting pathological traits. In this study, we sought to explore the neural correlates of response inhibition in MDD and BPD, as well as the pathological personality trait domains contained in the ICD-11 personality disorder model. We measured functional brain activity underlying response inhibition on a Go/No-Go task using functional magnetic resonance imaging in 55 female participants recruited into three groups: MDD without comorbid BPD ($n = 16$), MDD and comorbid BPD ($n = 18$), and controls with neither disorder ($n = 21$). Whereas response-inhibition-related activation was observed bilaterally in frontoparietal cognitive control regions across groups, there were no group differences in activation or significant associations between activation in regions-of-interest and pathological personality traits. The findings highlight potential shared neurobiological substrates across diagnoses and suggest that the associations between individual differences in neural activation and pathological personality traits may be small in magnitude. Sufficiently powered studies are needed to elucidate the associations between the functional neural correlates of response inhibition and pathological personality trait domains.

## Introduction

Cognitive control refers to the processes involved in generating and executing plans and goals, which support flexibility of thoughts and behaviours, adaptive behavioural responses, and

**Funding:** This work was supported by a Toronto Neuroimaging Institute Stimulus Grant, a Frederick Banting and Charles Best Canada Graduate Scholarship Doctoral Award (GSD-152335 [to DC]), an Early Researcher Award (ER14-10-185 [to ACR]) from the Province of Ontario's Ministry of Research and Innovation and a Research Excellence Faculty Scholar Award (to ACR) from the University of Toronto Scarborough, and a Discovery Grant from the Natural Sciences and Engineering Research Council of Canada (2018-04844 [to ACHL]). The funders had no role in study design, data collection and analysis, decision to publish, or preparation of the manuscript.

**Competing interests:** The authors have declared that no competing interests exist.

goal-directed thinking. Cognitive control is a superordinate construct that comprises more narrowly delineated abilities (sometimes referred to as "executive functions"), including attentional and inhibitory control, working memory, and cognitive flexibility [1]. Disruptions in cognitive control systems are linked to impulsive and risky behaviours, including potentially harmful actions, such as substance abuse [2], suicidal thoughts and suicide attempts [3] and disrupted eating behaviours [4], as well as so-called "externalizing" behaviours, more broadly [5].

Beyond reflecting a risk for impulsive behaviours, deficits in cognitive control are also associated with a range of psychiatric disorders, including mood disorders (such as major depressive disorder (MDD) [6]) and personality disorders (especially borderline personality disorder (BPD) [7, 8]), as well as many other psychiatric diagnoses [9]. Disturbances in cognitive control are linked to both the affective regulation [10] and impulse control components of MDD and BPD. Although not featured as a symptom in the diagnostic criteria for MDD, this disorder is associated with impulsive traits that are present not only during depressive mood states [11, 12] but also during periods of remission from depressive episodes [13]. Impulsivity is included in the diagnostic criteria for BPD, comprising such behaviours as excessive spending, risky sexual activity, substance abuse, reckless driving, and binge eating [14]. Identifying the factors that underpin impulsive behaviours in MDD and BPD—including neurobiological variables—is crucial for understanding the risk for potentially damaging actions in people with these diagnoses.

Given that MDD and BPD appear to share impulsivity as one—among potentially other—underlying symptom dimensions, neurobiological systems subserving cognitive control might jointly underlie these disorders. However, little research has compared cognitive control processes between MDD and BPD, and no study to our knowledge has investigated the neural correlates of cognitive control across these diagnoses. Furthermore, the neural correlates of cognitive control processes have not yet been investigated in relation to pathological personality trait dimensions most associated with impulsive behaviours, namely, disinhibition and negative affectivity [15, 16]. In this exploratory study, we examined the neural correlates of a subcomponent of cognitive control—specifically, response inhibition—across MDD and BPD and explore the association of these neural correlates with pathological personality traits that cut across these diagnoses.

## Response inhibition

Response inhibition refers to the ability to inhibit impulses and dominant or habitual behavioural responses, allowing for the selection of more appropriate behaviours for a given goal [17]. Similarly, response inhibition may be conceptualized as one's ability to exert self-control in a moment of action. Various behavioural tasks have been developed to assess response inhibition, including Go/No-Go (GNG), Stop-Signal (SST), and Stroop tasks. These behavioural tasks have in common the requirement that examinees exert cognitive control over their actions or responses to a certain stimulus or class of stimuli. The present thesis focuses on the GNG task because it represents one of the most extensively studied measures of response inhibition [18] and has very commonly been used in neuroimaging research [19–22]. Conventional GNG tasks normally incorporate two types of stimuli: Go and No-Go stimuli. These stimuli are typically presented as visual or auditory cues. Examinees are asked to respond to Go stimuli via a behavioural response (e.g., a manual button press) and to withhold that response to No-Go stimuli. In neuroimaging studies, a "blocked design" is frequently employed, which includes groupings (or blocks) of trials comprising different proportions of Go and No-Go stimuli. "Go" blocks are normally constituted entirely of Go stimuli, whereas

"No-Go" blocks feature a mix of both Go and No-Go stimuli (although the exact proportions vary across studies [19]). This form of task design develops a habituation to supplying a behavioural response to Go stimuli, such that when No-Go stimuli are presented, examinees must exert cognitive control to prevent responding in a habituated fashion. GNG task instructions are relatively straightforward for examinees to understand, and variations of the task are commonly applied in psychopathology research [18].

GNG tasks have been used in behavioural studies of MDD and BPD to investigate response inhibition as one subcomponent of cognitive control. As one example in MDD, Kaiser et al. [23] used an auditory GNG task with two difficulty conditions (low and high) based on pitch disparity, wherein the low difficulty pitch was perfectly discriminable (100% recognition rate) and the high difficulty pitch was less discriminable (80% recognition rate). During the Go condition, the participants responded to the infrequent stimuli but ignored the frequently occurring stimuli, whereas in the No-Go condition, participants responded to the more frequent stimuli and inhibited their responses to the less frequent No-Go stimuli. Compared to healthy controls (HC), MDD participants made more commission errors but only in the No-Go condition, and no differences in reaction times (RT) were observed for any of the conditions or groups. This suggests that poorer performance in MDD was due to lower response inhibition and not related to specific elements of the task (e.g., difficulty or spurious noise interference).

A separate line of behavioural studies of response inhibition in MDD has incorporated affective components (e.g., emotional words or faces) into the conventional GNG task, a modification that is thought to capture the interface between affect and cognitive control [24, 25]. While not the focus of the present study, these so-called "emotional" GNG tasks further illuminate response inhibition processes in MDD. Participants with MDD perform more poorly than HC on behavioural performance measures, including making more commissions and omissions, and either slowed or hastened RT patterns. This disparity in performance is seen to an even greater degree when the tasks involve the processing of affective stimuli, such as emotionally charged faces [26, 27] or emotional words [28].

Individuals with BPD have also been shown to perform below HCs on GNG tasks across various behavioural outcome measures. Rentrop et al. [29] investigated an auditory GNG task and compared BPD and HC participants. The task was also set up into two difficulty conditions ("simple" and "difficult") where the difference between tone pitch influenced the detection difficulty. Difficulty level of the task was associated with a slight but significant increase in RT across the entire sample and significantly lower performance for all participants; however, there were no interactions with participant group. Participants with BPD made more commission errors than HC on No-Go trials. Koudys and Ruocco [30] obtained results consistent with these findings in a family study of BPD that included first-degree biological relatives. Several measures of executive function were administered to participants, including a continuous performance test that permitted an examination of the frequency of commission errors to infrequently presented stimuli. Participants with BPD made more commission errors than both relatives and HC, and in a discriminant function analysis, the combination of both lower response inhibition and interference resolution significantly differentiated participants with BPD from the other two groups.

However, findings of lower response inhibition performance in BPD have not been observed in all studies. Hagenhoff et al. [31] sought to investigate potential deficits in executive functions (including working memory and response inhibition) in participants with BPD compared to controls. Response inhibition was assessed with a GNG task in which participants were shown pictures of shapes and were instructed to respond or not respond to either a triangle or square shape, depending on the assigned target stimuli, with No-Go blocks consisting of

75% Go stimuli. The researchers found no significant group differences in omissions, commissions, or RT.

It is also relevant to note that participants with BPD have shown lower performance compared to HC on an emotional GNG task [32]. Sinke et al. [32] used an emotional GNG task in which participants were asked to respond to specific geometric figures as target stimuli and to inhibit their responses to other figures. There were also three emotion conditions—neutral, happy, and angry faces—as background stimuli. BPD showed lower $d$-prime ($d$') and slower RTs to Go trials relative to HC in both the happy and angry conditions. The authors suggest that the lower $d$' and slower RTs may relate to a difficulty in assessing and interpreting social signals in an efficient way, or that it could also reflect key differences in the cognitive control functioning of patients with BPD.

## Neural correlates of response inhibition

Interest in how the brain subserves response inhibition originally stems from studies of patients with a brain lesion, insult, or form of neural degeneration (e.g., traumatic brain injury, tumour, stroke, epilepsy, or dementia) affecting the right hemisphere who appeared to be particularly prone to exhibiting disinhibited behaviours [33–35]. Lesions in multiple brain regions have been implicated in deficits in patients' abilities to inhibit behaviours appropriately, including the right orbitofrontal cortex, ventromedial frontal cortex, inferior frontal gyrus (IFG) and middle frontal gyrus (MFG), insula, subregions of the temporal cortex, right parietal cortex, and temporal-parietal junction [33, 35, 36]. Given the wide range of brain regions that seem to be related to a variety of disinhibited behaviours, neuroimaging studies have tended to focus on cognitive tasks that assess response inhibition to uncover brain regions that might underlie disinhibited behaviour.

**Controls.** Functional neuroimaging results from control samples show that response inhibition tasks commonly elicit greater activation in the superior, middle, and inferior frontal gyri, as well as regions such as the insula, cingulate cortex, and the parietal lobule and temporoparietal junction [21]. Somewhat mirroring what is observed in studies of patients with brain lesions, neuroimaging research on response inhibition tasks tend to reveal more activation lateralized to the right hemisphere [21, 37], suggesting that the functioning of regions in the right hemisphere is strongly implicated in response inhibition.

A meta-analysis by Zhang, Geng and Lee [38] synthesized the results of 225 studies to examine the common and differential neural correlates of response inhibition-based cognitive processes (i.e., tasks of interference resolution, action withholding, and action cancellation). They also sought to determine if the dominance of the right frontal cortical regions in response inhibition was maintained across these studies, or if a different network explanation was supported. Overall, they found that there was considerable common activation of right hemisphere regions, including the IFG, insula, medial cingulate, paracingulate gyri, and the superior parietal gyrus. When examining network level activation, they found that the engagement of the frontoparietal and ventral attentional networks was fundamental in the response inhibition process, and that differential activation in these networks could be distinguished for different subprocesses of response inhibition. Taken together, these findings provide a foundation for interpreting the results of neuroimaging research on cognitive control in MDD and BPD, which could illuminate subcomponents of response inhibition processes that may be disturbed in these disorders.

**MDD.** There is a paucity of neuroimaging studies examining response inhibition in MDD, although it is notable that these studies include participants from across the lifespan (for an overview of these studies, see Table 1). Additionally, the studies vary according to other

**Table 1. Neuroimaging studies of response inhibition in major depressive disorder.**

| Authors | Imaging-Task Type | Affectivity | Participants | Primary Findings |
|---|---|---|---|---|
| Bobb Jr et al., 2012 [44] | fMRI-SST | Non-Affective | 15 Late Life Depression patients 13 HC | No performance differences were found between the groups. Depressed patients showed greater activation relative to controls in the left-lateralized frontostriatal-limbic circuitry, including the bilateral superior frontal cortices, left orbitofrontal gyri, left insular cortex, left cingulate cortex, left caudate and left putamen. Controls did not show this increased activation. Suggests that depressed group requires this additional activation to perform similarly to controls. |
| Eugène et al., 2010 [45] | fMRI-Custom Response Inhibition Task | Affective | 12 MDD patients, 12 never-depressed controls | Depressed participants increased activation in the rostral anterior cingulate cortex (rACC) during inhibition of negative words. In controls this activation was associated with inhibition of positive words. |
| Ho et al., 2017 [43] | fMRI-GNG | Non-Affective | 45 Adolescent MDD patients, 53 matched HC | MDD patients showed inflexibility in the anterior cingulate cortex node of the Central Executive Network, the right dorsal anterior cingulate cortex (dACC) and medial frontal gyrus. Individual flexibility of the right dACC and medial frontal gyrus predicted response inhibition performance. |
| Korgaonkar et al., 2013 [39] | fMRI-GNG | Non-Affective | 30 Outpatients with MDD, 30 matched HC | MDD patients showed hyperactivation of the dorsomedial PFC during response inhibition tasks relative to controls. |
| Langenecker et al., 2007 [40] | fMRI-Parametric GNG | Non-Affective | 20 MDD patients, 22 matched HC | MDD patients showed lower behavioural performance than HC. MDD patients also showed a slower reaction time. MDD patients showed greater activation of the frontal and anterior temporal areas during correct rejections, compared to HC. Suggesting a possible compensatory mechanism. Response inhibition activation in the bilateral inferior frontal and left amygdala, left insula and left nucleus accumbens predicted antidepressant treatment outcomes. |
| Ruchsow et al., 2008 [41] | EEG-GNG | Non-Affective | 21 MDD patients in partial remission, 21 matched HC | MDD patients showed a significantly reduced Nogo-P3 amplitude relative to controls. |

MDD = Major Depressive Disorder; HC = Healthy Controls; PFC = prefrontal cortex; fMRI = Functional Magnetic Resonance Imaging; GNG = Go/No-Go; SST = Stop Signal Task; EEG = Electroencephalography.

factors, such as the imaging technique, specific response inhibition task, and whether the investigation occurred as part of a clinical trial (including comparing pre- and post-treatment effects).

Using functional magnetic resonance imaging (fMRI), Korgaonkar et al. [39] compared outpatients with MDD to HC on a GNG task and found that the former group showed greater activation of the anterior cingulate cortex (ACC) and lower activation of the right dorsolateral prefrontal cortex (dlPFC) during response inhibition on their GNG task relative to the latter group. Behaviourally, MDD was found to be significantly slower than controls on their GNG task, however they were found to perform with similar accuracy as the HC.

Langenecker et al. [40] used a parametric GNG task, which included three levels of difficulty ranging from a sustained response task to build a prepotent responding pattern to target letters, to two progressively more difficult versions of GNG, which at the second level required participants to respond to target letters based on their appearance pattern, and at the third level, additional targets to the initial target set were added. They found that the MDD group performed below HC, showing slowed reaction times relative to HC and more omissions than HC. In the context of the MDD group showing slower RTs and more omissions than HC, the former group also showed greater activation in the frontal and anterior temporal areas during No-Go Trials. These findings were interpreted to suggest that greater activation may be required for participants with MDD to perform similarly to HC, suggesting a possible compensatory mechanism.

Response inhibition studies in MDD have also used other neuroimaging methods and tasks. Ruchsow et al. [41] used electroencephalography (EEG) and showed that MDD patients in partial remission have a significantly lower No-Go-related P3 amplitude relative to HC. The P3 component is a centroparietal related positivity thought to be associated with frontoparietal brain region activity. Lower P3 amplitudes are associated with internalizing and externalizing psychopathology [42], the former of which is most relevant to MDD. Ho et al. [43] employed a modified response inhibition task with adolescents with MDD and HC and using a network approach found that those with MDD showed inflexibility in the efficiency of the right dorsal ACC and medial frontal gyrus (MedFG). Furthermore, individual differences in the efficiency of these regions were associated with successful response inhibition on the behavioural task. A study of individuals with late-life depression [44] used a SST, which asks participants to cancel a motor response that has already been initiated, which may enhance inhibitory demands compared to response inhibition alone. The depressed group showed greater activation of the left frontostriatal-limbic circuitry, composed of the bilateral superior frontal cortices and left orbitofrontal gyrus, insular cortex, cingulate cortex, caudate and putamen, while HC did not show greater activation in these areas.

As previously mentioned, research on MDD has also used emotional GNG tasks, which can provide insights into the interface between affect and response inhibition. Emotional stimuli are often thought to be more relevant or salient to depressed individuals, due to the higher levels of negative affect typically seen in MDD. For example, on a response inhibition task incorporating negative words, Eugène et al. [45] found that MDD showed higher activation of the right ACC when inhibiting negative words, while HC instead demonstrated higher activation in this region to positive words. Additionally, MDD showed greater activation of the left putamen during the inhibition of negative words. These findings suggest that, at least in some cases, cognitive control processes may only be disrupted when stimuli or situations incorporate emotional materials, which may reflect key cognitive disturbances in this disorder.

**BPD.** Several neuroimaging studies have investigated the neural correlates of response inhibition in BPD (see Table 2 for an overview of these studies). Mortensen et al. [46] used fMRI and a GNG task to compare BPD and HC. They found that both groups showed similar patterns of brain activation, but that higher self-reported impulsive behaviours were correlated with less activation in the left orbitofrontal cortex, left amygdala, left precuneus, and bilateral cingulate cortices in both BPD and HC. This might suggest that individual differences in impulsive behaviours are more strongly associated with response inhibition-related brain activity than the comparison between BPD and HC groups. Similarly, van Eijk et al. [47] used a mix of response inhibition tasks (i.e., Simon, GNG, and SST) and found no significant differences in behavioural performance or brain activation between BPD and HC.

Wrege et al. [48] used a novel GNG task with oddball trials to control for attentional saliency when studying response inhibition in BPD and HC. During No-Go trials compared to Go trials, BPD showed lower activation in the right posterior frontal gyrus, MFG, and IFG. They also found that individuals with BPD showed less activation of the left-hemispheric inferior frontal junction, frontal pole, superior parietal lobe, and supramarginal gyrus for oddball compared to Go trials. When associating brain activity with symptom reports, they found that higher levels of activation in the ventral lateral PFC (vlPFC) for No-Go compared to Go trials was associated with greater self-rated BPD symptoms. Additionally, lower vlPFC activation for oddball compared to Go trials was associated with higher self-reported ratings of attentional impulsivity. These results suggest that individuals with BPD show disrupted activation within the frontoparietal network and that the activity differences are likely associated with their level of symptoms, including impulsiveness.

**Table 2. Neuroimaging studies of response inhibition in borderline personality disorder.**

| Authors | Imaging-Task Type | Affectivity | Participants | BPD Patient Comorbidity | Primary Findings |
|---------|-------------------|-------------|--------------|------------------------|------------------|
| Albert et al., 2019 [50] | EEG-GNG | Non-Affective | 20 BPD patients, 20 matched HC | Excluded any history of brain trauma or neurological disease, substance use disorders or dependence within the last year, and previous bipolar or psychotic diagnosis. Included current comorbid dysthymia, panic disorder, bulimia nervosa. | BPD patients showed more commission errors than controls. Event Related Potentials (ERP) showed that both groups displayed greater frontocentral P3 amplitude for response inhibition. Source reconstruction revealed that BPD patients also active posterior parietal regions to aid in response inhibition, while controls activated prefrontal regions. |
| Jacob et al., 2013 [53] | fMRI-GNG | Affective | 17 female BPD patients, 18 matched HC | Excluded current psychotropic medication other than SSRIs, excluded current depression or substance dependence, lifetime schizophrenia or bipolar disorder. | During anger induction, HC showed increased activity of the left Inferior Frontal Cortex during successful inhibition trials. BPD patients instead showed an increase in the subthalamic nucleus. Suggests compensatory recruitment of differential brain regions in BPD. |
| Mortensen, Rasmussen, & Håberg, 2010 [46] | fMRI-GNG | Non-Affective | 15 female BPD patients, 15 matched HC | Excluded anyone with a history of psychiatric disorder other than BPD. | BPD patients had more commission errors in the No-Go blocks. BPD group had higher ratings for impulsivity, sensitivity to punishment and sensitivity to reward. Impulsivity scores correlated negatively with activation of the left orbitofrontal cortex, left amygdala, left precuneus, bilateral cingulate cortices during response inhibition for everyone. Sensitivity to punishment correlated negatively with activation of the right superior frontal gyrus and parahippocampal gyrus. |
| Ruchsow et al., 2008 [51] | EEG-Flanker/GNG Hybrid | Non-Affective | 17 BPD patients, 17 matched HC | N/A | BPD patients showed reduced Nogo-P3 amplitude relative to controls. (Nogo-P3 is related to response inhibition and response conflict.) |
| Ruocco et al., 2021 [49] | fNIRS-GNG | Non-Affective | 86 BPD patients, 60 non-affected first-degree relatives, 83 controls | Excluded lifetime diagnosis of psychotic disorders, current substance use disorders, neurological or medical illnesses that could impact brain function. | BPD patients showed bilateral decreases in PFC activation during response inhibition in comparison to their relatives and to controls. |
| Silbersweig et al., 2007 [56] | fMRI-GNG | Affective | 16 BPD patients, 14 matched HC | Excluded current medical and neurological conditions or substance dependences, | BPD patients showed decreased ventromedial prefrontal cortical activity to response inhibition of negative words, relative to controls. |
| Soloff et al., 2017 [54] | fMRI-GNG | Affective | 31 female BPD patients, 25 matched HC | Excluded lifetime diagnosis of schizophrenia, delusional (paranoid) disorder, bipolar disorder or psychotic depression and current substance use disorders. | The amygdala exerted greater modulatory control over its target regions in BPD patients during negative affective condition. Controls showed higher activation of the right orbitofrontal cortex, right dorsal anterior cingulate cortex, right parietal cortex, right basal ganglia and right dorsolateral PFC. |
| van Eijk et al., 2015 [47] | fMRI-Simon, GNG, SST and a Hybrid Response Inhibition Task | Non-Affective | 44 female BPD patients, 43 matched HC | Excluded lifetime diagnoses of ADHD, Schizophrenia or Bipolar Disorder, Substance Use Disorders within last 3 years and current depressive episode. | Found no difference in behavioural performances. Found no significant differences in brain activation patterns during any tasks. Concluded that in emotionally neutral conditions, response inhibition is not impaired in patients with BPD, without co-occurring ADHD. |

(*Continued*)

**Table 2.** (Continued)

| Authors | Imaging-Task Type | Affectivity | Participants | BPD Patient Comorbidity | Primary Findings |
|---|---|---|---|---|---|
| van Zutphen et al., 2020 [94] | fMRI-GNG | Affective | 53 BPD patients, 20 clinical patient controls, 34 controls | Excluded lifetime diagnosis of psychotic disorder, bipolar disorder, dissociative identity disorder, active substance use disorders. | BPD patients showed more omission errors than controls to all categories (i.e., neutral, negative). BPD showed more activation in the inferior parietal lobule and frontal eye fields when inhibiting responses to negative stimuli compared to neutral stimuli. Inferior parietal lobule activation correlated with the impulsivity subscale of the BPD checklist. |
| Wingenfeld et al., 2009 [55] | fMRI-Stroop | Affective | 20 BPD patients, 20 matched HC | High rates of comorbidity in the BPD group, including PTSD, depressive disorders, social phobia, bulimia nervosa, somatoform pain disorder. | BPD patients had overall slower reaction times in the Stroop task compared to HC. Unrelated to emotional interference. HCs showed more activation in the ACC and Frontal Cortex. BPD patients did not show any equivalent signal changes, suggesting they are not again to engage these regions for regulating stress and emotions in the Stroop task. |
| Wrege et al., 2021 [48] | fMRI-Oddball GNG | Non-Affective | 45 BPD patients, 29 HC | Excluded any patients with psychotic symptoms. Included comorbidities of depressive disorders, anxiety disorders, eating disorders, somatoform disorders, PTSD, substance use disorders. | Individuals with BPD showed lower activation in the right posterior frontal gyrus, MFG, IFG, during No-Go trials compared to Go trials. BPD showed less activation of the left-hemispheric inferior frontal junction, frontal pole, superior parietal lobe and supramarginal gyrus for oddballs compared to Go trials. Found higher levels of activation in the ventral lateral PFC for No-go compared to Go was associated with greater self-rated BPD symptoms. BPD patients show disrupted activation of the frontoparietal network in response inhibition. |

BPD = Borderline Personality Disorder; ADHD = Attention Deficit Hyperactivity Disorder; HC = Healthy Controls; ACC = Anterior Cingulate Cortex; PFC = prefrontal cortex; fMRI = Functional Magnetic Resonance Imaging; GNG = Go/No-Go; SST = Stop Signal Task; EEG = Electroencephalography.

In a study of participants with BPD, their first-degree biological relatives, and HC, functional near-infrared spectroscopy (fNIRS) was used to measure PFC activity during a GNG task [49]. BPD participants showed bilateral decreased activation of the anterior portions of the PFC during response inhibition compared to both relatives and HC.

Albert et al. [50] used EEG and source localization and found that participants with BPD recruit different brain regions for response inhibition compared to HC on a GNG task. They found that BPD relied more on activation of the posterior parietal regions when inhibiting responses, while HC recruited more prefrontal regions. In another EEG study, Ruchsow et al. [51] showed that BPD patients showed a reduced No-Go P3 amplitude relative to controls on a flanker-GNG hybrid task.

Like MDD research, some BPD studies have also used emotional GNG tasks, which could illuminate the neural underpinning of the interaction between emotion and behavioural control in BPD, which is thought to be especially relevant to this disorder [52]. Jacob et al. [53] used fMRI and an affective manipulation in their GNG task. They found that if anger was

induced before completing a GNG task, HC showed higher activity of the left inferior frontal cortex during successful inhibition trials, while individuals with BPD did not show this increase, and instead showed an increase in the subthalamic nucleus. They suggest that this might be a key element in how individuals with BPD compensate for a lack of activation in frontal cortical regions during response inhibition. Similarly, Soloff et al. [54] found that the amygdala exerted greater modulatory control during a negative affective condition of a GNG task in BPD patients. Soloff et al. also showed that controls had higher activation in the right lateralized regions that are traditionally thought to support response inhibition, suggesting that participants with BPD might ineffectively recruit brain regions on these tasks. Wingenfeld et al. [55] used an affective Stroop task to illicit inhibitory control-related brain activation. In response to negative emotional words, HC showed more significant activation in the ACC, as well as the MedFG and the precentral gyrus, while the BPD group did not show this increased activation. This further supports the idea that participants with BPD might have difficulty recruiting frontal cortical regions involved in inhibitory processing, albeit within the context of affective stimuli. Finally, using an emotional GNG task, Silbersweig et al. [56] observed that lower ventromedial PFC (vmPFC) activity was associated with behavioural inhibition of negative words in BPD relative to controls. They also found that lower levels of constraint and higher negative emotion were associated with this lower vmPFC activity and with higher activity of amygdalar-ventral striatal regions during inhibition of negative words in BPD.

## Dimensional classification of personality psychopathology

In recent years, proposals have been put forward to advance current conceptualizations of psychopathology (including personality pathology), with major implications for authoritative psychiatric diagnostic nosologies [57]. In the field of personality disorders, there are many proponents of a dimensional system of diagnosis [58] that has seen traction in both the Alternative Model for Personality Disorders (AMPD) in Section III of the Diagnostic and Statistical Manual of Mental Disorders—Fifth Edition (DSM-5) [14] and the proposal that appears in the International Statistical Classification of Diseases and Related Health Problems—Eleventh Revision (ICD-11) [59]. These dimensional classifications are considered a significant advancement because they appear to improve on many of the limitations of categorical diagnoses, which include high rates of diagnostic comorbidity and low validity and reliability [57]. For example, MDD and BPD are highly comorbid, with rates potentially as high as 80% [60]. Alternative hierarchical and dimensional systems of psychiatric classification might explain the high degree of diagnostic comorbidity between MDD and BPD, for example, based on shared symptom dimensions and perhaps underlying cognitive and neurobiological systems [61]. Importantly, categorical diagnoses often see a loss of information due to the collapsing of individuals—who might show a range of symptomatology—into a single diagnostic group [57]. A dimensional approach allows for the retention of these individual differences (e.g., in pathological personality traits) and may better describe the underlying trait and symptom dimensions, while explaining much of the comorbidity across related categorical diagnoses.

In both the DSM-5 AMPD and the ICD-11 model, personality disorder is defined by an impairment in personality functioning, defined as disturbances in self and/or interpersonal functioning, and one or more pathological personality trait dimensions [14, 59]. According to the AMPD, these traits comprise Negative Affectivity, Detachment, Antagonism, Disinhibition, and Psychoticism. The ICD-11 model includes Negative Affectivity, Detachment, Dissociality, Disinhibition, and Anankastia. Negative Affectivity and Disinhibition are traits common to both the AMPD and ICD-11 models.

In the AMPD, Disinhibition is described as the pathological counterpart to trait conscientiousness and has several facets, including irresponsibility, impulsivity, distractibility, risk taking, and (a lack of) rigid perfectionism [14]. Similarly, in the ICD-11 model, Disinhibition is described as manifesting in the forms of impulsivity, distractibility, irresponsibility, recklessness, and a lack of planning [59]. In both the AMPD and ICD-11 models, higher levels of disinhibition are reflected in behaviours that are oriented towards immediate gratification, and where an individual would be more likely to act on impulse derived from their current feelings, thoughts, or reactions to stimuli in their environment, and without considerations of possible negative consequences.

Negative Affectivity is described in the AMPD as relating to Neuroticism, and is characterized by the presence of emotional lability, anxiousness, tendency towards depression, hostility and altered relationship patterns (e.g., separation insecurity, submissiveness) [14]. The ICD-11 describes Negative Affectivity as the tendency to experience negative emotions broadly, which are more frequent and intense (i.e., out of proportion) than a given situation would call for [59]. The ICD-11 description of Negative Affectivity also highlights the presence of emotional lability, poor emotional regulation, low self-esteem and confidence and mistrustfulness. In both models, the tendency to experience negative emotions in general, and to have higher lability in those emotions are present. This may lead an individual to be highly depressed, anxious, or irritable and to have difficulty controlling their emotions and reactions to situations.

Both Negative Affectivity and Disinhibition are especially relevant to both MDD and BPD. These traits are also thought to be two components of what constitute the larger construct of impulsivity: Joyner et al. [62] make a case that impulsivity is composed of externalizing problems and impulsive traits, such as Negative Urgency (NU). In their model, externalizing problems are driven almost entirely by Disinhibition, where "impulsigenic traits" (e.g., NU) are dominated by Negative Affectivity. Settles et al. [63] showed that NU is related to various externalizing behaviours, such as alcohol abuse and dependence, smoker status, aggression, risky sexual behaviour, illicit drug use, and conduct disordered behaviour. They also suggest that externalizing behaviours may be underpinned by an affective laden decision, which requires processing of emotional information.

Negative Affectivity and Disinhibition may also represent transdiagnostic pathological personality traits that cut across MDD and BPD, although the neurobiological underpinnings associated with these traits across these disorders have yet to be investigated. However, neuroimaging research on closely related symptom and trait dimensions provides some indication of the patterns of brain activation that may be associated with these pathological traits. The neural basis of Neuroticism, a personality trait conceptually tied to Negative Affectivity [57], has been extensively studied. In a meta-analysis of neuroimaging studies investigating Neuroticism, the trait is broadly related to activity in areas such as the ACC, thalamus, hippocampus and parahippocampus, striatum, as well as various frontal, temporal, parietal, and occipital lobe subregions [64]. Functional neuroimaging studies reviewed in this meta-analysis primarily investigated aspects of Neuroticism, such as fear learning, anticipation of negative stimuli, and emotional processing. The implicated regions may be related to and crucial in our understanding of the neural basis of Negative Affectivity. Similarly, regions identified in Servaas et al. [64] as being key in response inhibition research, such as the supramarginal gyrus and ACC, also overlap with the regions associated with Neuroticism.

Neuroimaging research relevant to Disinhibition has mainly focused on the Externalizing spectrum [65], which includes symptoms such as impulsivity, substance-use problems, aggression, and attentional problems [57]. Externalizing has been associated with functional connectivity in key brain networks, such as lower connectivity in the frontoparietal and dorsal attentional networks [66] and greater connectivity between the salience and ventral attentional

networks [67]. Regions such as the insula, striatum, ACC, medial and lateral PFC, amygdala and hippocampus are also related to Externalizing [65, 67–69]. Notably, there is considerable overlap among the brain regions identified in neuroimaging research on Neuroticism, Externalizing, and response inhibition, which may suggest common neural substrates underlying the relationships of Negative Affectivity and Disinhibition with response inhibition, and perhaps cognitive control, more broadly.

## The present study

Building on prior neuroimaging research examining response inhibition in MDD and BPD, the present exploratory study compares neural activation during response inhibition among the following participant groups: (a) participants with comorbid diagnoses of MDD and BPD, (b) participants with MDD without a comorbid diagnosis of BPD, and (c) controls with neither diagnosis. The primary aim of this exploratory investigation is to examine potential shared and distinct neural activation patterns underlying response inhibition between MDD and BPD, including whether any disruptions in neural activation are specific to one diagnosis compared to the other. Given that the participant groups are also expected to vary according to the corresponding severity of pathological personality traits, a secondary aim of this work is to explore the extent to which individual differences in pathological personality trait dimensions—especially, Negative Affectivity and Disinhibition—relate to response inhibition-related neural activation. The latter aim could provide initial evidence of the neurobiological basis of these pathological personality trait dimensions as they relate to neuroimaging-based biomarkers relevant to cognitive control.

Given prior neuroimaging findings using the GNG task in control samples and participants with either or both diagnoses, both the MDD and BPD groups were hypothesized to show lower activation in right frontal cortical regions, right insula, right cingulate, and right parietal regions, when compared to controls. There were no specific expectations regarding differences between the MDD and BPD groups in response-inhibition-related brain activity because research has yet to directly compare these groups. To examine potential group differences, traditional, whole-brain activation statistical analyses were applied and supplemented by region-of-interest (ROI) analyses and context-dependent psychophysiological interactions (gPPI), which permitted an investigation of functional interactions between brain regions. To investigate the secondary aim, we used whole-brain activation statistical analyses as described above, using the individual differences (i.e., pathological personality trait scores) as the variable of interest. Given the previously proposed connection between Negative Affectivity and Disinhibition, and the available neuroimaging results, we anticipated that these traits would be significantly associated with response inhibition-related neural activation in the right frontal cortical regions, insula, cingulate and parietal regions, and that participants endorsing higher levels of both of these traits (regardless of participant group) would show less activation in the same regions compared to those lower in both traits.

## Materials and methods

### Participant characteristics

The study was approved by the Social Sciences, Humanities and Education Research Ethics Board at the University of Toronto. Written consent was obtained from participants. Sixty-four female adults (mean$_{age}$ = 28.73 years, SD$_{age}$ = 9.01) were recruited from the Greater Toronto Area as part of a larger study on biomarkers of depression and personality traits (see Gardhouse et al. and Carcone et al. for additional information on sample comorbidities, nicotine use, caffeine use, and other detailed procedural instructions not relevant to the present

study [70, 71]). Participants were recruited via advertisements posted online and in local clinics. All participants were right-handed, English speakers. Only females were recruited for the study due to the potential confounding variable of biological sex on the neuroimaging variables [72] and the expectation (based on past research in the same laboratory using similar recruitment strategies [8]) that very few males with BPD would be recruited to the study.

Other psychiatric diagnoses (e.g., Depressive Disorders, Anxiety Disorders, Obsessive Compulsive and Related Disorders, Trauma and Stressor Related Disorders) were permitted with the exception of the following: lifetime diagnosis of any DSM-5 psychotic or bipolar (I or II) disorders, neurodevelopmental disorders (e.g., autism-spectrum disorder), any neurological illnesses, moderate or severe traumatic brain injuries, serious physical illness that would have significant implications for brain structure or function (e.g., myocardial infarction, cancer), or a current (i.e., past three months) DSM-5 diagnosis of a substance use disorder or eating disorder, as well as pregnancy, lactation, or any use of anti-inflammatory drugs.

Prospective participants completed the McLean Screening Instrument for BPD (MSI-BPD) [73] upon telephone screening to determine whether they had a likely BPD diagnosis. From the initially recruited 64 participants, 40 presented with a current major depressive episode and 20 of these individuals' scores on the MSI-BPD suggested likely BPD. The design of the present study permitted an investigation of what might be unique to BPD relative to MDD by methodologically controlling for MDD across the two MDD participant groups (e.g., MDD alone versus MDD with comorbid BPD). The remaining 24 individuals did not report a current major depressive episode and did not produce a score on the MSI-BPD to suggest the potential presence of BPD.

After initial intake, three participants were identified as having a current substance use disorder and were excluded from further analyses. Additionally, two participants were excluded from the study due to brain pathology, two participants had MRI contraindications (i.e., metal in the body), and one participant could not tolerate the scanning environment and did not complete the task. GNG performance scores were calculated using $d'$ by z-transforming both the hit rate and false alarm rate, and then subtracting the false alarm rate from the hit rate. Due to the task being designed to measure successful response inhibition by achieving high levels of accuracy, 34 participants had perfect accuracy scores that were corrected using the loglinear adjustment approach from Hautus [74] in order for the scores to be used for the calculation of $d'$. Following the calculation of accuracy scores, one additional participant was excluded from further analysis due to a negative $d'$ score indicating an exceptionally poor performance that would suggest inadequate compliance with task instructions. The final sample included 55 participants (mean$_{age}$ = 28.51 years, SD$_{age}$ = 8.47): 18 participants with comorbid MDD and BPD, 16 participants with MDD without comorbid BPD, and 21 controls with neither diagnosis. The groups were found to not significantly differ in terms of age, $F(2,52) = 0.66$, $p = .52$, $\omega^2 = -.013$.

## Procedures

All participants completed the Structured Clinical Interview for DSM-5 [75] conducted by a senior clinical psychology doctoral student under the supervision of a licensed psychologist with expertise in the diagnosis of MDD and BPD. Participants also completed the personality inventory for ICD-11(PiCD) to assess levels of pathological personality traits (including Negative Affectivity and Disinhibition) [76]. Other measures not relevant to the present study are described in Gardhouse et al. and Carcone et al. [70, 71].

Neuroimaging data were collected using a Siemens Prisma 3T Full-Body MRI Scanner at the Toronto Neuroimaging Facility (https://toni.psych.utoronto.ca). Functional data were

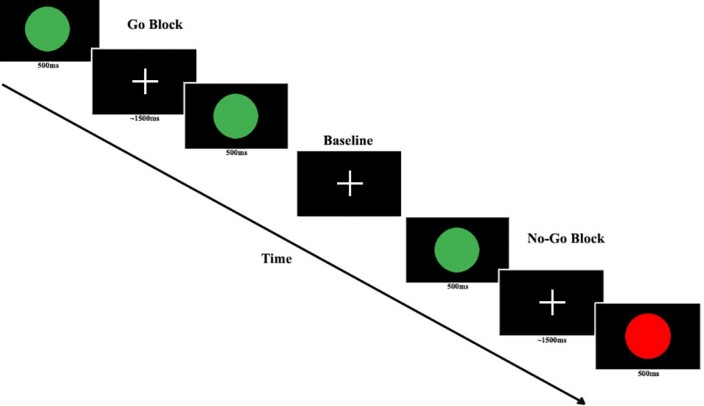

**Fig 1. Graphical depiction of the design of the Go/No-Go task.**

acquired over two identical acquisition runs using a T2*-weighted Blood Oxygen Level Dependent (BOLD) echo-planar imaging (EPI) sequence (69 interleaved oblique slices per volume; 142 volumes; TR = 2000 ms; TE = 31 ms; voxel size = 2 x 2 x 2 mm; FOV = 220; matrix size = 110 x 110; FA = 70˚). A high-resolution 3D anatomical scan was acquired for each participant using a T1-weighted MPRAGE sequence (160 slices; TR = 2000 ms; TE = 2.4 ms; voxel size = 1 x 1 x 1 mm; FOV = 256; matrix size = 256 x 256; FA = 9˚). Finally, field maps were acquired using a dual gradient echo sequence to allow for correction of magnetic field inhomogeneities (69 slices; TR = 679 ms; $TE_1$ = 4.92 ms; $TE_2$ = 7.38 ms; voxel size = 2 x 2 x 2 mm; FOV = 192; matrix size = 96 x 96; FA = 60˚).

Participants completed a visual GNG task across two EPI runs, with the option for a break in between the runs. Each run had four Go blocks and four No-Go blocks, with the No-Go blocks having an equal proportion of Go and No-Go trials. The blocks were presented in a constant alternating fashion (i.e., Go, No-Go, Go, No-Go, and so forth). Within the No-Go blocks, trials were presented in a pseudo-randomised fashion, where there would never be more than two No-Go trials in a row. In total across both runs together, 144 Go trials and 48 No-Go trials were presented. Before the task, participants were instructed to attend to the screen and respond by pressing a button on a keyboard to all green circles (i.e., Go Stimuli) and not to respond to red circles (i.e., No-Go stimuli). Stimuli were presented for 500ms with an average jittered interstimulus interval (ISI) of 1500 ms (Fig 1). Sessions ran for an average of 12 min, 23 s (range: 10 min, 26 s– 15 min, 21 s).

## Statistical power

Besides the number of participants, several factors determine the univariate statistical power of an fMRI study, including the number of task trials, the temporal spacing of the trials, and individual differences in MRI signal-to-noise ratio. Notably, the number of participants included in the present study is comparable to that in previous neuroimaging studies of MDD and BPD examining between-group differences [40, 41, 44–46, 53, 55]. Furthermore, the design of the GNG task in the present study has advantages over prior neuroimaging investigations of MDD and BPD by featuring two acquisition runs and a greater number of GNG task trials, which together increase the statistical power of the present work. The secondary analyses examining associations between neural activation and pathological personality traits are exploratory, and while the sample size of the present study only provides sufficient statistical power to detect

large effects, these exploratory analyses nevertheless provide estimates of the strengths of these effects that can inform future work.

## Statistical analyses

Statistical analyses for behavioural data were conducted using IBM SPSS Statistics Version 28 [77]. Normality and other assumptions were tested using appropriate visual and statistical methods, as suggested by Field [78]. No transformations were required and all relevant assumptions were met. Group differences in demographic characteristics, personality traits, and behavioural performance were tested using ANOVA with bootstrapped confidence intervals (1000 samples) and post-hoc tests using Bonferroni corrections for multiple comparisons. We report $\omega^2$ effect sizes alongside ANOVA results, where a small effect is .01, a medium effect is .06 and a large effect is .14 [78].

Imaging data were preprocessed and subsequently analyzed using FEAT (FMRI Expert Analysis Tool) Version 5.98 and other packages from FSL (FMRIB software library; http://www.fmrib.ox.ac.uk/fsl) [79]. Participant data were first individually visually checked for any major distortions or movement noise. Data were then corrected for motion using MCFLIRT [80] and B0 unwarping was applied to correct for distortions resulting from magnetic field inhomogeneities. Functional data were also temporally filtered with a 90-s high pass filter and spatially smoothed using a Gaussian kernel of 5-mm full-width half-maxima. Individual functional data were co-registered to their respective high resolution 3D anatomical brain scan and normalized to Montreal Neurological Institute space (MNI-152) using a combination of linear and non-linear transformations with FLIRT and FNIRT [80, 81]. Finally, an independent component analysis was conducted on the data acquired from each functional run using MELODIC [82] to isolate noise components that were identified by their spatial profile, time-course, and power spectrum. These noise components were removed prior to statistical analysis.

Each run of the GNG task for each participant was put into a general linear model (GLM) with the different events being specified as explanatory variables (EVs) or predictors in the model and then convolved with a double-gamma model of the hemodynamic response function. The first-level GLM included three EVs in total, with each including all images and both EPI runs within the sequence. These EVs included Go, No-Go, and Baseline. The first-level GLM included 12 contrasts composed of every possible comparison of the three EVs (e.g., No-Go>Go, No-Go>[Baseline+Go], etc.). The first-level GLM resulted in one parameter estimate image being created for each EV and contrast, for each run and each participant.

Both individual runs for each participant were then entered into a second-level fixed effects analysis and the resulting statistical images (contrast of parameter estimate [COPE] maps) were subsequently entered into a higher-level group analysis for the examination of group mean effects and group differences. These second-level COPE maps were also entered into additional higher-level dimensional analyses for each of the PiCD pathological personality traits. Statistical inference was conducted using a non-parametric approach as implemented by the Randomise tool (http://www.fmrib.ox.ac.uk/fsl/randomise) in conjunction with threshold-free cluster-enhancement (TFCE) [83]. TFCE identifies clusters of activity without pre-determining an arbitrary cluster-defining threshold and using permutation testing, which can achieve a multi-threshold meta-analysis of the random field theory cluster $p$-values to determine statistical significance. We used 10000 permutations for statistical inference, using a corrected family-wise threshold of $p < 0.05$.

In addition to whole-brain analysis, ROI analyses were performed to examine response inhibition-related functional brain activity. Selection of ROIs for this study was informed by

the GNG relevant clusters obtained from a seminal meta-analysis [21]. The top three clusters were selected, which created a ROI mask composed of the right insula, MFG, IFG, inferior parietal lobule, and MedFG, as selected from the Harvard-Oxford Structural Atlas. To complement the ROI mask based on Swick et al. [21] and to be exhaustive, an additional exploratory ROI mask was constructed. The additional mask was composed of the significant task-based activity found when collapsing response-inhibition-related activity across all participants.

A gPPI analysis was conducted using FSL to examine task-dependent changes in functional connectivity. Time series data were first extracted from a seed voxel selected based on the most significant clusters of activation identified by the No-Go>Go contrast in the whole-brain univariate analysis. This was then submitted to an independent gPPI analysis as follows. A first-level GLM was implemented with the following task conditions and variables specified as EVs: EV1: Go; EV2: No-Go; EV3: Baseline; EV4: Time series of seed voxel; EV5: Go x Time series of seed voxel; EV6: No-Go x Time series of seed voxel; and EV7: Baseline x Time series of seed voxel. All possible contrasts between EV5, EV6 and EV7 were also specified to identify any differential patterns of functional connectivity between task conditions. At the second level, a fixed effects analysis was then conducted by combining the runs for each participant, and finally, group level inference was completed using the randomise tool in conjunction with TFCE.

## Results

### Participant characteristics and behavioural performance

Table 3 displays participants' demographic characteristics, PiCD scores, and behavioural performance. ANOVA with 1000 bootstrapped samples and post-hoc (Bonferroni) comparisons revealed that participant groups did not differ significantly with regard to behavioural

**Table 3. Demographic characteristics, pathological personality trait scores, and behavioural performance on the Go/No-Go Task.**

| | Control ($n$ = 21) Mean (SD) | MDD ($n$ = 16) Mean (SD) | BPD and comorbid MDD ($n$ = 18) Mean (SD) | F(df) | Post-hoc group differences[a] |
|---|---|---|---|---|---|
| Age, years | 27.71 (8.28) | 30.56 (9.35) | 27.61 (8.02) | 0.66 (2) | n.s. |
| Age Range, years | 18–48 | 19–47 | 18–46 | | |
| *PiCD[b] Scores* | | | | | |
| Negative Affectivity | 33.43 (8.52) | 39.75 (6.65) | 45.67 (6.38) | 13.49 (2) | BPD > Controls***, MDD > Controls* |
| Detachment | 27.90 (6.94) | 28.75 (7.70) | 32.89 (9.04) | 2.13 (2) | n.s. |
| Dissocial | 21.95 (5.45) | 22.38 (5.10) | 32.94 (8.74) | 16.13 (2) | BPD > MDD***/Controls*** |
| Disinhibition | 23.95 (7.99) | 27.75 (7.89) | 33.72 (10.84) | 5.75 (2) | BPD > Controls** |
| Anankastic | 40.90 (7.56) | 41.31 (7.92) | 37.17 (7.88) | 1.55 (2) | n.s. |
| PiCD Total Score | 148.14 (19.71) | 159.94 (18.85) | 182.39 (20.68) | 14.74 (2) | BPD > MDD**/Controls*** |
| *Go/No-Go Task* | | | | | |
| Commissions | 1.24 (1.58) | 1.56 (1.50) | 2.50 (2.68) | 2.04 (2) | n.s. |
| Omissions | 3.52 (5.22) | 3.50 (6.77) | 4.11 (7.73) | 0.05 (2) | n.s. |
| True Negatives | 46.76 (1.58) | 46.44 (1.50) | 44.17 (5.91) | 2.83 (2) | n.s. |
| True Positives | 140.48 (5.22) | 138.98 (11.23) | 135.89 (17.69) | 1.02 (2) | n.s. |
| Reaction Time, ms | 352.03 (61.77) | 329.76 (39.34) | 334.67 (63.15) | 0.81 (2) | n.s. |

[a] *p < .05, **p < .01, ***p < .001. Bonferroni correction.

[b] PiCD = Personality Inventory for ICD-11.

performance on commissions, $F(2,52) = 2.04$, $p = .14$, $\omega^2 = .036$, omissions, $F(2,52) = 0.05$, $p = .95$, $\omega^2 = -.036$, true negatives, $F(2,52) = 2.83$, $p = .07$, $\omega^2 = .063$, true positives, $F(2,52) = 1.02$, $p = .37$, $\omega^2 = .001$, or correct Go trials RT, $F(2,52) = 0.81$, $p = .45$, $\omega^2 = -.007$. Regarding pathological personality traits, participant groups differed in Disinhibition, $F(2,52) = 5.75$, $p = .006$, $\omega^2 = .147$ where BPD showed higher levels than controls (95% CI [2.62, 16.92]), Negative Affectivity, $F(2,52) = 13.49$, $p < .001$, $\omega^2 = .312$, where both BPD (95% CI [6.40, 18.08]) and MDD (95% CI [0.29, 12.35]) were higher than controls, and Dissocial, $F(2,52) = 16.13$, $p < .001$, $\omega^2 = .355$, where BPD showed higher levels than both MDD (95% CI [4.94, 16.20]) and controls (95% CI [5.73, 16.26]). The groups did not differ on Detachment, $F(2,52) = 2.13$, $p = .13$, $\omega^2 = .039$, or Anankastia, $F(2,52) = 1.55$, $p = .22$, $\omega^2 = .020$.

### Neuroimaging results

**Whole brain analyses.** When contrasting BOLD responses on No-Go trials with Go trials ('No-Go>Go contrast'), significant clusters of activation were observed bilaterally in the superior frontal gyri (SFG), MFG and precentral gyri, frontal orbital cortex, putamen, caudate, postcentral gyrus, inferior parietal lobule, supramarginal gyrus, superior temporal gyrus, superior parietal lobule, and precuneus. Left lateralized activation was also detected in the IFG and the superior temporal gyrus, while right lateralized activation was also found in the thalamus (Fig 2). Non-parametric statistical testing at the whole-brain level revealed similar patterns of activity for this contrast (see Table 4).

Whole-brain analyses of Go activity (Go>Baseline contrast) and No-Go activity (No-Go>Baseline contrast) were conducted to compare participant groups and no statistically significant differences were observed. There were also no significant differences even when exploring the data further using a more liberal statistical threshold of p < .001 uncorrected. Similarly, analyses comparing participant groups in response-inhibition-related brain activity (No-Go>Go contrast) revealed no statistically significant differences. The same result was obtained when BPD and MDD were collapsed into a single "patient" group and compared to controls. Furthermore, regressing individual differences in pathological personality traits across all participants (regardless of group) onto response-inhibition-related brain activity at the whole-brain level revealed no statistically significant associations.

**Region-of-interest and trait-based analyses.** Analyses of ROIs (i.e., ROIs informed by Swick et al. [21] and task-based activity were applied to the present analyses) across participant groups on response-inhibition-related activity revealed no significant group differences in both corrected and uncorrected analyses as described above. Additionally, ROI analyses were conducted to regress pathological personality traits onto response-inhibition-related activity and there were no significant results. To estimate the strengths of the associations between neural activity and each of the pathological personality traits, mean activity values were extracted from response-inhibition-related regions implicated in our results for the No-Go>Go contrast and correlated with PiCD scores. The results are presented in Table 5 and reveal Pearson correlations ranging from a minimum of .00 to a maximum of |.26| (between Disinhibition and the right frontal pole). No correlations were statistically significant.

**Generalized psychophysiological interaction analysis.** A peak voxel from the right MFG (x = 32, y = 42, z = 40, p = .01) was selected for use in extracting the time series data. Our gPPI analysis found that there were no significant task-dependent changes in functional connectivity across the participant groups in both corrected and uncorrected analyses as described above.

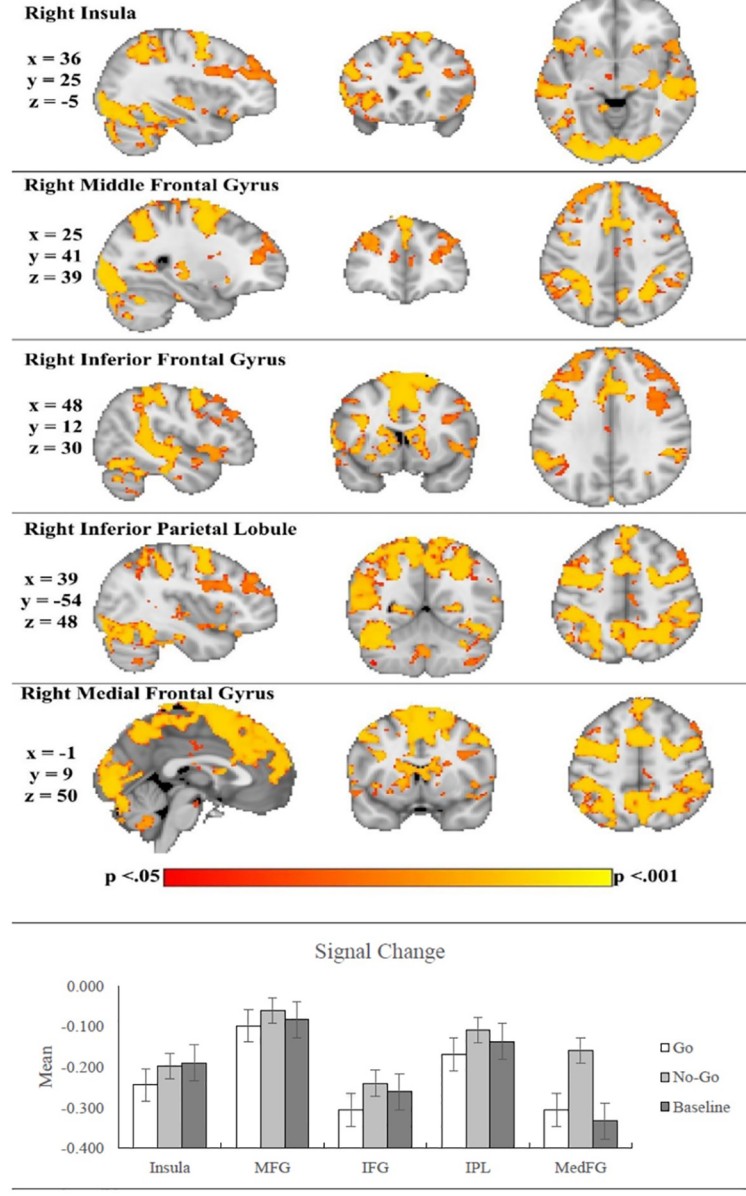

**Fig 2. Patterns of brain activation for No-Go>Go contrast collapsed across all participants.** X, y, z refer to MNI coordinates. Sagittal (left), Axial (center), and Horizontal (right) slice orientation, rendered on MNI152 2mm brain standard. Slice locations selected based on the top 5 brain regions from GNG analyses in Swick et al. [21]. Activation reflects significantly increased BOLD signal. Bar graph depicts signal change at the corresponding locations. Error bars are standard error. MFG = Middle Frontal Gyrus; IFG = Inferior Frontal Gyrus; IPL = Inferior Parietal Lobule; MedFG = Medial Frontal Gyrus.

## Discussion

The present exploratory study is the first to our knowledge to compare MDD and BPD in response-inhibition-related brain activation and the association of this activation with pathological personality traits. The primary hypothesis was not supported: participant groups did not significantly differ in response-inhibition-related brain activation, which was examined using both parametric and non-parametric statistical testing in whole-brain analyses.

**Table 4. Significant clusters of activity associated with the No-Go > Go contrast.**

| Region | X | Y | Z |
|---|---|---|---|
| **No-Go>Go Contrast** | | | |
| **Cluster 1–116334 voxels—p < .001** | | | |
| Right Middle Frontal Gyrus | 32 | -2 | 58 |
| Right Superior Frontal Gyrus | 10 | 12 | 70 |
| Left Superior Frontal Gyrus | -14 | -2 | 76 |
| Left Superior Temporal Gyrus | -50 | -38 | 6 |
| Lett Precentral Gyrus | -16 | -10 | 74 |
| Left Inferior Temporal Gyrus | -50 | -58 | -26 |
| **Cluster 2–31 voxels—p < .001** | | | |
| Right Parahippocampal Gyrus | 30 | 2 | -24 |
| **Cluster 3–20 voxels—p < .001** | | | |
| Left Cingulate Cortex | -10 | -42 | 22 |
| **Cluster 4–4 voxels—p < .001** | | | |
| Right Frontal Pole | 32 | 46 | 0 |

For Cluster 1, theoretically relevant local maxima are reported; Coordinates are presented in MNI space.

Similarly, no significant group differences were observed in ROI or functional connectivity-based analyses. Contrary to expectations, pathological personality traits were not significantly related to response-inhibition-related brain activation in whole-brain and ROI analyses. Examining the patterns of correlations with ROI's revealed small-to-medium effects mainly involving the right insula, right IFG, and right MFG. These results were obtained in the context of no significant group differences in behavioural performance, which was expected based on the design of the task to focus on successful response inhibition. Additionally, higher levels of Negative Affectivity and Disinhibition were observed in one or both patient participant groups ((i.e., MDD alone, and BPD with comorbid MDD) versus controls.

Although no differences in brain activity were revealed across the groups and no significant associations were observed with pathological personality traits, statistically significant activity was detected across all participants for the primary response inhibition contrast (i.e., No-Go>Go) in theoretically relevant brain regions. Specifically, regional patterns of brain activation were consistent with Swick et al.'s [21] meta-analysis, which revealed activity in the SFG,

**Table 5. Two-tailed Pearson correlations between pathological personality traits and peak voxel location for response inhibition-related neural activity.**

| Brain Region (x, y, z) [a] | Negative Affectivity | Detachment | Dissocial | Disinhibition | Anankastic | Total Score |
|---|---|---|---|---|---|---|
| Paracingulate Gyrus (-2, 10, 52) | -.006 | -.124 | -.018 | .053 | -.105 | -.062 |
| Left Superior Temporal Gyrus (-48, -38, 6) | .128 | .056 | .169 | .075 | -.171 | .098 |
| Right Frontal Pole (30, 44, 12) | .028 | .230 | .206 | .204 | -.205 | .173 |
| Right Inferior Frontal Gyrus (60, 16, 4) | .044 | .239 | -.057 | .121 | -.015 | .120 |
| Right Frontal Orbital Cortex (34, 28, -24) | -.118 | -.150 | .000 | .087 | -.178 | -.115 |
| Right Middle Frontal Gyrus (32, 42, 40) | -.081 | -.123 | .061 | .065 | -.102 | -.056 |
| Right Frontal Pole (30, 50, 34) | -.163 | .110 | .073 | -.261 | .125 | -.063 |
| Right Superior Parietal Lobule (44, -34, 46) | -.113 | -.001 | -.048 | .193 | -.218 | -.051 |
| Right Precentral Gyrus (40, 0, 46) | -.082 | -.169 | -.062 | .021 | -.029 | -.108 |

Voxels from regions theoretically implicated in response inhibition were selected. Brain regions identified using the Harvard-Oxford Structural Atlas.

[a] Voxel coordinates presented in MNI space.

MFG, inferior parietal lobule, middle temporal gyrus, supramarginal gyrus, and precuneus. This suggests that although the task was low in difficulty (i.e., focused primarily on successful response inhibition), it nevertheless engaged regions typically implicated in response inhibition processes.

No significant differences were observed in whole-brain analyses when comparing participant groups. Collapsing MDD and BPD into a single group did not reveal any differences relative to controls. These findings are consistent with a small set of studies that have also found no differences involving these groups when examining response-inhibition-related brain activity [47]. However, the findings are inconsistent with studies that have detected group differences between MDD and controls in terms of activation in the ACC and PFC [39, 43], frontal and anterior temporal regions [40] and other frontolimbic regions [44]. Similarly, the present findings are also inconsistent with some studies comparing BPD and controls, which obtained differences in activation of key frontolimbic regions [46], ACC and frontal gyri clusters [48, 55], and anterior PFC [49, 56]. Additional analyses focused on ROI-based activity and gPPI analyses comparing the participant groups revealed found no significant differences. As all such analyses converged on similar findings, the results appear internally consistent and suggest that the study was likely underpowered to detect group differences due to the limited sample size. Although similar studies observed significant results with sample sizes in the same range as the present study (for example, see_Korgaonkar et al. [39], Ho et al. [43], Jacob et al. [53], or Soloff et al. [54]), those studies often compared two groups (rather than three, as was carried out in this work). Therefore, as previously stated, the lower power of the present study limited the detection of any potential group differences.

A secondary aim of the study was to explore the association of pathological personality traits with response-inhibition-related brain activity. Negative Affectivity and Disinhibition were of primary interest because they are most strongly related to MDD and BPD, respectively, which was supported by the results of group comparisons of these traits wherein BPD and MDD scored significantly higher in Negative Affectivity than controls, and BPD additionally scored significantly higher in Disinhibition than the controls. Unexpectedly, no significant relationships were uncovered between these traits and brain activity; however, the study was powered to detect large effects. Therefore, to illuminate the magnitudes of these potential effects in this exploratory study, the relationship between the traits and key regions previously implicated in response inhibition was examined. The observed correlations between pathological personality traits and mean activity within key response-inhibition-related brain regions were mostly in the small range. The highest correlation was between Anankastic and the right IFG ($r = -.27$), whereas the strongest associations for Negative Affectivity and Disinhibition were with the right MFG ($r = -.11$) and right IFG ($r = .14$), respectively.

It is possible that the smaller effect sizes observed in the exploratory correlational analyses are related to the range and extent of activation elicited by the GNG task, which focused on successful response inhibition. Although the GNG task elicited expected patterns of brain activity across the sample, it might be that the difficulty level of the task was not sufficiently challenging to ultimately detect differences between the groups. Approaches used in other studies—such as generating higher levels of difficulty by making the Go and No-Go stimuli more difficult to discriminate (e.g., changes in pitch disparity; see Kaiser et al., [23] and Rentrop et al., [29]) or incorporating affective stimuli [53, 55]—may be better suited for studying response inhibition in MDD and BPD. The present results may provide additional support to the idea put forward by van Eijk et al. [47] that response inhibition is mainly impacted in BPD when there is either an affective component to the task or other factors influencing cognitive control are present (e.g., attention-deficit hyperactivity disorder). This may also be the case for MDD, considering the overlap between BPD and MDD. However, Schulz et al. [25] provided

some evidence that affective and non-affective GNG tasks are effectively equivalent in terms of response inhibition performance outcomes. It may be that the addition of an affective component might only magnify any performance differences that are already present due to other general cognitive control factors. It is also possible that other components of response inhibition or additional aspects of cognitive control are affected in MDD and/or BPD, and potentially associated with Negative Affectivity and Disinhibition. Further research is needed to clarify these questions and the present study provides essential information about the potential magnitude of such effects, which can inform future work.

While GNG tasks provide a route for investigating response inhibition, it is important to recognize that these tasks also require participants to engage multiple other cognitive functions. Task demands require the execution of motor commands responsible for the action of responding to Go stimuli, and the inhibition of a motor response in the face of No-Go stimuli. They also require the relevant sensory processing (e.g., visual or auditory) to allow for the discrimination between the presented stimuli. It is also likely that other cognitive functions are involved, such as boundary decision making [84], lexical decision making [85], and any other functions that might be related to the unique demands of a given GNG design. This would presumably be reflected in differences in activation levels and patterns of brain activation relevant to these cognitive functions, such as Broca's area and Wernicke's area for lexical decision making [86], left fusiform and right superior temporal gyrus for visual processing [87], and the inferior frontal gyrus for motor commands [88]. These considerations must be kept in mind when interpreting the results of the present study, as they may impact the associations of brain activation with the diagnoses of interest and pathological personality traits. Despite the limitations of the task, the GNG remains one of the most researched tasks for studying response inhibition.

Although the results of the present exploratory study are not conclusive, the biological investigation of pathological personality traits as proposed in alternative dimensional models of personality disorder is an emerging area of inquiry. This is the first study to our knowledge to examine the relationship of these pathological personality traits to response-inhibition-related brain activation. The preliminary findings indicate that Negative Affectivity and Disinhibition—traits especially relevant to MDD and BPD—may be associated with the right frontal gyri cluster (i.e., MFG, IFG). However, future work with sufficiently large samples will be required to further investigate this emerging pattern of associations. In this context, it is crucial to consider that studies relating brain imaging to psychopathology symptoms likely require sample sizes numbering into the thousands to be adequately powered to produce meaningful and replicable results [89]. The present study contributes new information about the magnitude of the associations between response-inhibition-related brain activity and more recently conceptualized pathological personality traits, implying that the magnitudes of such associations may be similar to those identified in Marek et al. These findings also broadly support the process of validating key components of alternative dimensional models of personality disorder, which may be informative as these models are considered for adoption in authoritative diagnostic nosologies.

In addition to the previously discussed limitations of sample size a further limitation is that the task design included solely low- versus high-difficulty conditions, and more research is needed to uncover patterns of neural activation for moderate-difficulty conditions and associations with BPD and MDD diagnoses, as well as pathological personality traits. It is also important to consider that the sample was composed of solely female participants, which limits the generalizability of the results. Future studies should seek to use this approach with a sample that includes male participants. It is also critical to recognize that the control group was not exclusive of individuals with a history of some forms of psychiatric illness. Although this

encouraged a range of severity in pathological personality traits and likely facilitated individual differences analyses, it may also have limited the detection of group differences involving the control group due to overlapping symptoms (and, potentially, underlying neurobiology). To investigate individual differences in pathological traits and their associations with brain activity underlying response inhibition, future research might wish to forgo the use of specific diagnoses altogether and instead recruit individuals with a range of severity of pathological traits (e.g., by consecutively recruiting patients from a personality disorders treatment clinic). Such an approach would allow for the recruitment of individuals with a range of diagnoses or even subthreshold symptom presentations, and would also provide an avenue for expanding transdiagnostic domains of neuroimaging research. It might also be of interest for future studies to investigate other domains of cognitive control and how they relate to these pathological personal traits, as the present study focused solely on response inhibition. Although not mentioned in the present thesis, studies have also focused on other subconstructs of cognitive control, such as goal selection, task switching, response selection, and performance monitoring, among participants with MDD [90], BPD [91, 92], and a range of other disorders [93]. Studies focusing on these other areas of cognitive control may also benefit from a similar trait-based approach as the one adopted in the present study. Finally, it also important to recognize that attention deficit hyperactivity disorder (ADHD) was not included as part of the structured diagnostic assessment of the participants in the present study. This limitation may pose further challenges for the interpretation of our results because the differences between impulsivity related to ADHD versus BPD are not yet well established [95]. Future research should consider how comorbid ADHD might play a role in response inhibition processes in BPD and related pathological personality traits.

## Conclusion

Current understanding of the neurobiological basis of cognitive control and its associations with pathological personality traits across various forms of psychopathology is limited. Extant research mainly focuses upon externalizing symptoms and different conceptualizations of impulsivity, whereas no previous studies have investigated neuroimaging-based biomarkers of the pathological personality trait domains proposed in the DSM-5 AMPD and ICD-11, let alone across both MDD and BPD diagnoses. The present exploratory study uncovered expected patterns of brain activation associated with response inhibition but detected no significant differences in behavioural performance or response inhibition-related brain activity across participant groups. Whereas no significant relationships were observed between pathological personality trait domains and response-inhibition-related brain activity, exploratory correlational analyses revealed mainly small associations, which may inform future work incorporating sufficiently large samples to detect such effects. Taken together, this research lays the groundwork for future neuroimaging research on response inhibition in MDD and BPD, as well as pathological personality trait dimensions proposed in alternative diagnostic frameworks for personality disorders.

## Author Contributions

**Conceptualization:** Dean Carcone, Andy C. H. Lee, Anthony C. Ruocco.

**Data curation:** Dean Carcone, Katherine Gardhouse.

**Formal analysis:** Cody Cane.

**Funding acquisition:** Dean Carcone, Andy C. H. Lee, Anthony C. Ruocco.

**Methodology:** Dean Carcone, Andy C. H. Lee, Anthony C. Ruocco.

**Project administration:** Dean Carcone, Katherine Gardhouse, Anthony C. Ruocco.

**Resources:** Andy C. H. Lee, Anthony C. Ruocco.

**Software:** Andy C. H. Lee.

**Supervision:** Andy C. H. Lee, Anthony C. Ruocco.

**Writing – original draft:** Cody Cane.

**Writing – review & editing:** Cody Cane, Dean Carcone, Katherine Gardhouse, Andy C. H. Lee, Anthony C. Ruocco.

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
