## [Decision Letter · Decision Letter 0]

30 Sep 2022

PONE-D-22-21209An Exploratory Study of Functional Brain Activation Underlying Cognitive Control in Major Depressive Disorder and Borderline Personality DisorderPLOS ONE

Dear Dr. Cane,

Thank you for submitting your manuscript to PLOS ONE. After careful consideration, we feel that it has merit but does not fully meet PLOS ONE’s publication criteria as it currently stands. Therefore, we invite you to submit a revised version of the manuscript that addresses the points raised during the review process.

We look forward to receiving your revised manuscript.

Kind regards,

Jyrki Ahveninen

Academic Editor

PLOS ONE

Journal Requirements:

  "This work was supported by a Toronto Neuroimaging Institute Stimulus Grant, a Frederick Banting and Charles Best Canada Graduate Scholarship Doctoral Award (GSD-152335 [to DC]), an Early Researcher Award (ER14-10-185 [to ACR]) from the Province of Ontario’s Ministry of Research and Innovation and a Research Excellence Faculty Scholar Award (to ACR) from the University of Toronto Scarborough, and a Discovery Grant from the Natural Sciences and Engineering Research Council of Canada (2018-04844 [to ACHL])."

Reviewers' comments:

Reviewer's Responses to Questions

**Comments to the Author**

1. Is the manuscript technically sound, and do the data support the conclusions?

Reviewer #1: Yes

Reviewer #2: Yes

2. Has the statistical analysis been performed appropriately and rigorously? 

Reviewer #1: I Don't Know

Reviewer #2: Yes

3. Have the authors made all data underlying the findings in their manuscript fully available?

Reviewer #1: No

Reviewer #2: Yes

4. Is the manuscript presented in an intelligible fashion and written in standard English?

Reviewer #1: Yes

Reviewer #2: Yes

5. Review Comments to the Author

Reviewer #1: Comments

The aim of this study was to explore the functional brain activation underlying cognitive control in major depressive disorder and borderline personality disorder. Functional magnetic resonance imaging was used to study this brain activity underlying cognitive control, especially response inhibition. Participant groups with major depressive disorder with and without comorbid borderline personality disorder, and controls with neither disorder, were compared. A secondary aim was to explore the extent to which individual differences in pathological personality trait dimensions relate to response inhibition-related neural activation. The results showed that there was observed response-inhibition-related activation bilaterally in frontoparietal cognitive control regions across groups, but there were no group differences in activation in regions-of-interest, and no significant associations between activation and pathological personality traits.

Introduction

Introduction is very informative and carefully written, but maybe even too detailed and extensive (little like textbook text). Needing some summarizing and combining in general.

For example, on page 6 there is too detailed presentation of one single study (Rentrop et al.). And the same problem also for example on page 19 with Wrege et al. (and Ruocco et al.).

For example, on page 22 there is interesting discussion, but maybe too wide (and without references).

Perhaps some (compact) introduction to MDD and BPD, as well as fMRI, are needed.

On page 13, about BPD: “Only a small number of neuroimaging studies…” – maybe not so “small number” when you see the overview (Table 2).

On page 25, the hypotheses could need some more reference ‘to prior findings’, at least in the end (on page 26).

Materials and methods

Neuropsychiatry, especially attention deficit hyperactivity disorder, can be essentially affecting the results in this study. It is important to consider this issue more, at least reporting it, for example on page 26.

It is also necessary to report the information of the participant’s psychotropic medication. And also tobacco and maybe coffee.

On page 26, “Other psychiatric diagnoses were permitted” -what are these diagnosis?

On page 27, how this current substance use and brain pathology were studied, and what are “34 extreme scores”?

On page 28, in the end; was there 144 and 48 trials in one run, or in both runs together? And “participants were instructed to … and respond” -how to respond?

On page 29, could add some more facts about these stimuli and first clearly state that this GNG task was visual task (page 28).

On page 29, Statistical analysis: “Normality was tested for all data…” – after this sentence could state what was the result (with normality). And “All other assumptions of the relevant statistical tests were met” -what do these mean?

On page 31, both runs were “combined into” – how?

Results

On page 32, “groups did not differ significantly with behavioural performance” -to add “behavioural performance” in the sentence, could make the issue more clear.

Table 3 (and Materials and methods): For Demographic Characteristics, is there only the Age? Also it would be interesting to have the BPD group, but the sample size is probably too small? (And the Table 3 is slightly hard to read. Maybe some interval after “Age”, and to clarify what is “Total”.)

On page 35, “contrasting BOLD responses” -how the contrasting was done? And “Non-parametric statistical testing” – what does this mean? And Table 4 – maybe need some more text about these findings in Results and also in these Table texts. For example -what are these clusters?

Also the text after Table 4 on the page 36 (and 37 & 39) could need more accurate reporting. And for example “participant groups” -more specifically, what groups? And “no … differences were observed” -maybe more accurately. And “using a more liberal statistical threshold…” -what does this mean?

On page 39, “diagnostic groups” -what groups? Terms “diagnostic / participant groups” vary.

Fig 2: Signal Change (MedFG) -could this be something important to report?

Discussion

On page 39, here wasn’t the group with ´pure BPD´ - maybe the first sentence “directly compare” is too strongly stated. And “groups” -term again vary.

On page 40, “Although similar studies” – could need some references, like on page 41 “… are most strongly related to MDD and MPD”, and “key regions previously implicated in”.

Limitations could be stated here more thoroughly, like BPD sample size and the fact that there were only female participants.

Maybe some summarizing is needed, for example on page 44, where the analysis is nice but quite extensive. And is the new topic “schizophrenia” needed here, in the end?

(Also, here could mention some targets of this kind of research, for example to develop treatments for BPD and MDD. But this is not necessary).

Reviewer #2: Thank you for the opportunity to review this manuscript investigating neural correlates of cognitive control (inhibition) in major depressive disorder (MDD) and borderline personality disorder (BPD).

The authors used a GNG task presented during fMRI to compare response inhibition-related neural activation of three groups: Control vs. MDD vs. BPD + MDD. Groups were also compared on ICD personality traits, which were subsequently correlated with inhibition-related neural activity across the whole sample. Interestingly, there were no between groups differences on inhibition-related brain activity (including ROI and functional connectivity analyses).

This paper has a number of strengths. The theoretical and empirical rationale for the study is very clear and review of previous literature is comprehensive and helpful (e.g., response inhibition performance in MDD and BPD; neural correlates in healthy controls, MDD, and BPD ; dimensional classification of personality functioning). The methods and statical analysis are rigorous and sophisticated. Despite the lack of significant findings, this is an important and innovative study that forms a solid basis for future work in this emerging area. Recruitment of a clinical sample including fMRI methodology is challenging and the authors should be commended for their efforts, although sample size is not ideal. The explanations regarding lack of significant findings are clear and well considered, including considerations around power and task-difficulty. I would like to see some more detail regarding authors’ choice of a low-difficulty compared to high-difficulty inhibition task. Despite this, the authors rightly acknowledge that the current study has the potential to inform future work in this area and there are some interesting suggestions to consider in the design of future studies regarding a recruitment strategy focused on pathological traits (vs. traditional diagnoses) and transdiagnostic mechanisms underlying clinical severity across diagnostic groups.

Minor suggestions

Title

-I wonder whether it may be useful to replace “cognitive control” with “response inhibition” to be more specific

Abstract

-I appreciate the limits on word count, but abstract may benefit from including group sample sizes, and also a sentence or two around possible explanations for the lack of between-group differences and implications/future directions

Introduction

-Tables 1 and 2 are informative and very useful. It may be beneficial to also include year and country of publication for this high-level summary

-Page 8: Could you be more specific regarding the use of term “Non-patients”: Do you refer to healthy controls only? This may be a better term, and is used in other places throughout manuscript (note a minor inconsistency as the term is written as “nonpatient controls” – without hyphen on p. 7.)

Method

-Page 28: PiCD does not seem to be introduced and defined before first use here. Please add this information

Results

-Table 3: Typo “Personality Prait Scores”

Discussion

-Page 40: Typo “that have detected groups differences”

6. PLOS authors have the option to publish the peer review history of their article (what does this mean?). If published, this will include your full peer review and any attached files.

Reviewer #1: No

Reviewer #2: **Yes: **Ely Marceau

---

## [Author Response · Author response to Decision Letter 0]

9 Nov 2022

We want to thank you for the opportunity to revise our submitted manuscript. We have taken the opportunity to address the editor’s and reviewers’ comments as well as to ensure that our manuscript meets PLOS ONE’s style requirements.

We would like to make changes to our financial disclosure. We would like to amend the statement as follows:

"This work was supported by a Toronto Neuroimaging Institute Stimulus Grant, a Frederick Banting and Charles Best Canada Graduate Scholarship Doctoral Award (GSD-152335 [to DC]), an Early Researcher Award (ER14-10-185 [to ACR]) from the Province of Ontario’s Ministry of Research and Innovation and a Research Excellence Faculty Scholar Award (to ACR) from the University of Toronto Scarborough, and a Discovery Grant from the Natural Sciences and Engineering Research Council of Canada (2018-04844 [to ACHL]). The funders had no role in study design, data collection and analysis, decision to publish, or preparation of the manuscript."

We have also updated our data availability, to allow it to be made public and meet PLOS ONE’s data availability requirements. We have uploaded a minimal anonymized dataset to a public repository (OSF), and we would like to provide the link and a DOI to the dataset:

https://osf.io/u6yxf/

doi: 10.17605/OSF.IO/U6YXF

Reviewer 1: Introduction is very informative and carefully written, but maybe even too detailed and extensive (little like textbook text). Needing some summarizing and combining in general. For example, on page 6 there is too detailed presentation of one single study (Rentrop et al.). And the same problem also for example on page 19 with Wrege et al. (and Ruocco et al.). For example, on page 22 there is interesting discussion, but maybe too wide (and without references). Perhaps some (compact) introduction to MDD and BPD, as well as fMRI, are needed. On page 13, about BPD: “Only a small number of neuroimaging studies…” – maybe not so “small number” when you see the overview (Table 2). On page 25, the hypotheses could need some more reference ‘to prior findings’, at least in the end (on page 26).

Response: We would like to thank the reviewer for their feedback. We agree that summarizing this information will be more concise and streamline the literature review. Accordingly, we have made the recommended changes (see p. 6 and p. 19). We also addressed the comment regarding references in the review of the dimensional traits on page 22. Similarly, on page 13, we have changed the quoted passage to read as follows:

 “Several neuroimaging studies…”

Finally, regarding the suggestion about our hypotheses, we have additionally added clarifying text to directly tie our hypotheses to the review of the relevant literature presented in the introduction (see pp. 25-26).

Reviewer 1: Neuropsychiatry, especially attention deficit hyperactivity disorder, can be essentially affecting the results in this study. It is important to consider this issue more, at least reporting it, for example on page 26. It is also necessary to report the information of the participant’s psychotropic medication. And also tobacco and maybe coffee. 

Response: We appreciate the reviewer’s valuable comments and apologize for not initially addressing these issues. While we agree that ADHD may impact our findings, we unfortunately did not systematically assess the diagnosis in this study. Considering the potential influence of the diagnosis, we acknowledge the issue in our limitations section (see pp. 44-45):

“Finally, it also important to recognize that attention deficit hyperactivity disorder (ADHD) was not included as part of the structured diagnostic assessment of the participants in the present study. This limitation may pose further challenges for the interpretation of our results because the differences between impulsivity related to ADHD versus BPD are not yet well established [95]. Future research should consider how comorbid ADHD might play a role in response inhibition processes in BPD and related pathological personality traits.”

95. Ditrich I, Philipsen A, Matthies S. Borderline personality disorder (BPD) and attention deficit hyperactivity disorder (ADHD) revisited–a review-update on common grounds and subtle distinctions. Borderline Personality Disorder and Emotion Dysregulation. 2021 Dec;8(1):1-2. doi: 10.1186/s40479-021-00162-w

Nicotine and caffeine use patterns were also not formally gathered; however, prior to the MRI scan, participants were asked to fast for a minimum of 8 hrs prior, as well as to abstain from the use of drugs or alcohol for two days before testing. They were also asked to avoid anti-inflammatory medications and intense physical exercise for 24 hrs before the study participation and to aim for at least 8 hours of sleep the night before. Additional information about psychotropic medication use can be found within Gardhouse et al. [70] and Carcone et al. [71]. Therefore, we have added information to the methods section (see p. 26) to make this additional information more clearly accessible to interested readers:

“see Gardhouse et al. and Carcone et al. for additional information on sample comorbidities, nicotine use, caffeine use, and other detailed procedural instructions not relevant to the present study [70-71]”

Reviewer 1: On page 26, “Other psychiatric diagnoses were permitted” -what are these diagnosis? On page 27, how this current substance use and brain pathology were studied, and what are “34 extreme scores”?

Response: We apologize for the confusion and are glad to clarify that outside of our explicitly excluded diagnoses as indicated in the Methods section, other comorbid diagnoses were permitted (e.g., anxiety disorder, PTSD, and OCD). We have added DSM category headers as clarifying text on page 26:

“Other psychiatric diagnoses (e.g., Depressive Disorders, Anxiety Disorders, Obsessive Compulsive and Related Disorders, Trauma and Stressor Related Disorders) were permitted…”

Current substance use disorders and brain injury were assessed using the Structured Clinical Interview for DSM (SCID) and other relevant questionnaires, as described in the procedure section of the manuscript (see pp. 26-27).

The extreme scores referred to perfect accuracy scores on the Go/No-Go behavioural task, which was completed during the fMRI scan. Perfect scores are not usable in the traditional calculation of d’. To clarify our language, we have changed this sentence as follows:

“34 participants had perfect accuracy scores that were corrected using the loglinear adjustment approach from Hautus [74] in order for the scores to be used for the calculation of d’.”

Reviewer 1: On page 28, in the end; was there 144 and 48 trials in one run, or in both runs together? And “participants were instructed to … and respond” -how to respond? On page 29, could add some more facts about these stimuli and first clearly state that this GNG task was visual task (page 28).

Response: The 144 and 48 trials are referring to both runs together. We have added this information to page 29 so that it is no longer ambiguous. We have also added information about how the participants were instructed to respond (using keyboard button presses). As suggested by the reviewer, we have also now stated explicitly that this is a visual task.

Reviewer 1: On page 29, Statistical analysis: “Normality was tested for all data…” – after this sentence could state what was the result (with normality). And “All other assumptions of the relevant statistical tests were met” -what do these mean?

Response: We would like to thank the reviewer for highlighting this point. As the language in the original was unclear, we have adjusted this text (see p. 30) and would like to clarify for the reviewer that the original intention of this information was to acknowledge that we conducted the appropriate tests of normality and checked the assumptions required for running our statistical analyses, as recommended in chapters 6 and 11 in Field [78]. 

To report all details of these tests in the manuscript would likely be unnecessary as there was nothing exceptional from these checks. To be specific about our methods though, we visually inspected the data using histograms and Q-Q plots where relevant. We also conducted tests using Levene’s test, as well as the Kolmogrov-Smirnov and Shapiro-Wilks. 

Reviewer 1: On page 31, both runs were “combined into” – how? On page 35, “contrasting BOLD responses” -how the contrasting was done? And “Non-parametric statistical testing” – what And Table 4 – maybe need some more text about these findings in Results and also in these Table texts. For example -what are these clusters? Fig 2: Signal Change (MedFG) -could this be something important to report?

Response: We appreciate the reviewer’s questions about our neuroimaging results. After reflecting on our language in the manuscript, we chose to change “combined” to “entered” throughout this section of the text to reduce ambiguity (see p. 31). This process is explained in the Methods section; briefly, we intend to communicate that the fixed effects analysis represents the mean of the two runs. We apologize if any of the following information was unclear: the contrasting of BOLD responses and the non-parametric methods are outlined in the Methods section with relevant references for any readers who would like to learn more about the technical details of the fMRI analyses that were conducted. Regarding the comments about Table 4 and Figure 2, these results largely speak for themselves and are just part of the standard reporting of these types of data and statistical analyses. The MedFG is also already listed as part of the larger contrast. These results are not of particular relevance to the present study’s hypotheses and aims so we respectfully suggest that any further explanation would likely be unnecessary.

Reviewer 1: On page 32, “groups did not differ significantly with behavioural performance” -to add “behavioural performance” in the sentence, could make the issue more clear. Table 3 (and Materials and methods): For Demographic Characteristics, is there only the Age? Also it would be interesting to have the BPD group, but the sample size is probably too small? (And the Table 3 is slightly hard to read. Maybe some interval after “Age”, and to clarify what is “Total”.) does this mean? 

Response: Regarding the comment about the text on page 32, we would like to thank the reviewer for this suggestion. We have revised the manuscript accordingly. In Table 3, only age that is relevant to the present study. As suggested, we have added the interval/range of ages to the table. We have also clarified what “Total Score” refers to, which is the total scale score of the Personality Inventory for ICD-11. Additionally, we apologize for the unclear presentation of the BPD group in Table 3. We would like to clarify that the “BPD+MDD” column header was referring to BPD with comorbid MDD. We have changed the column header to reflect this information. The MDD column header refers to participants that had MDD without BPD. The sample size of our BPD group (n=18) is larger than the MDD group (n=16). While our sample sizes are relatively small compared to many other studies (which we state in the Discussion section), the BPD group is only slightly smaller than the control group, and is slightly larger than the MDD group.

Reviewer 1: Also the text after Table 4 on the page 36 (and 37 & 39) could need more accurate reporting. And for example “participant groups” -more specifically, what groups? And “no … differences were observed” -maybe more accurately. And “using a more liberal statistical threshold…” -what does this mean? On page 39, “diagnostic groups” -what groups? Terms “diagnostic / participant groups” vary.

Response: We again thank the reviewer for pointing out these inconsistencies and ambiguities. To be consistent, we have changed all references of “diagnostic groups” to “participant groups” and clarified instances in which the manuscript is specifically referring to patients versus controls. The participant groups we refer to were originally outlined in our “Present Study” section of the Introduction (pp. 24-25). To clarify, when referring to the results of participant group comparisons, we compared across all three of the participant groups identified in the Methods section, and no statistically significant differences were observed across any of the group comparisons. Finally, the “more liberal statistical threshold” refers to the p-value threshold for the fMRI results (i.e., the more liberal threshold was p < 0.05).

Reviewer 1: On page 39, here wasn’t the group with ´pure BPD´ - maybe the first sentence “directly compare” is too strongly stated. And “groups” -term again vary. On page 40, “Although similar studies” – could need some references, like on page 41 “… are most strongly related to MDD and BPD”, and “key regions previously implicated in”.

Response: We would like to clarify that there was no “pure BPD” group. The BPD group included participants who had a diagnosis of BPD with comorbid MDD. We agree that the phrase “directly compare” may be too strong and have revised the manuscript to remove the word “directly”. We have also added relevant references as suggested by the reviewer (see p. 40).:

“(for example, see Korgaonkar et al. [39], Ho et al. [43], Jacob et al. [53], or Soloff et al. [54])”

Reviewer 1: Limitations could be stated here more thoroughly, like BPD sample size and the fact that there were only female participants. Maybe some summarizing is needed, for example on page 44, where the analysis is nice but quite extensive. And is the new topic “schizophrenia” needed here, in the end? (Also, here could mention some targets of this kind of research, for example to develop treatments for BPD and MDD. But this is not necessary).

Response: We appreciate the reviewer’s thorough reading of the manuscript and detailed suggestions. Accordingly, we have now adjusted the limitations section to state that only female participants were included in the study (p. 44):

“In addition to the limitations of sample size and features of the task design, it is also important to consider that the sample was composed of solely female participants, which limits the generalizability of the results. Future studies should seek to use this approach with a sample that includes male participants.”

Given that we have acknowledged our limited sample size throughout the Discussion section (including based on a previous comment), we did not repeat this item in the limitations section. We also agree that the comment in the Discussion about schizophrenia is irrelevant and we have removed it in our revised manuscript.

Reviewer 2: This paper has a number of strengths. The theoretical and empirical rationale for the study is very clear and review of previous literature is comprehensive and helpful (e.g., response inhibition performance in MDD and BPD; neural correlates in healthy controls, MDD, and BPD; dimensional classification of personality functioning). The methods and statical analysis are rigorous and sophisticated. Despite the lack of significant findings, this is an important and innovative study that forms a solid basis for future work in this emerging area. Recruitment of a clinical sample including fMRI methodology is challenging and the authors should be commended for their efforts, although sample size is not ideal. The explanations regarding lack of significant findings are clear and well considered, including considerations around power and task difficulty. I would like to see some more detail regarding authors’ choice of a low-difficulty compared to high-difficulty inhibition task. Despite this, the authors rightly acknowledge that the current study has the potential to inform future work in this area and there are some interesting suggestions to consider in the design of future studies regarding a recruitment strategy focused on pathological traits (vs. traditional diagnoses) and transdiagnostic mechanisms underlying clinical severity across diagnostic groups.

Response: We would like to thank the reviewer for their kind words and enthusiasm about this work. We are glad to provide additional insight into our choice of employing a low-difficulty compared to high-difficulty task: due to time constraints in the MRI scanning session, the Go/No-Go (GNG) task was designed in such a manner that allowed for a direct comparison between task conditions that would be likely to maximize brain activation differences while using as brief of a task as possible. As anticipated, the within-subjects comparison of the low-difficulty versus high-difficulty task conditions produced a robust pattern of response-inhibition-relevant brain activity, which corresponded with expected neural activation patterns based on a well-established scientific literature based on this task. We also viewed the task and broader research design as “exploratory” (as per the manuscript title) with the aim of evaluating the robustness of the task design for eliciting the hypothesized brain responses and estimating the strengths of the associations with the outcomes of interest. Of course, the reviewer’s comment is crucial to consider and should be a focus of future research on the topic. Accordingly, we have revised our limitations section as follows (p. 44):

“…a further limitation is that the task design included solely low- versus high-difficulty conditions, and more research is needed to uncover patterns of neural activation for moderate-difficulty conditions and associations with BPD and MDD diagnoses, as well as pathological personality traits.”

Reviewer 2: Title-I wonder whether it may be useful to replace “cognitive control” with “response inhibition” to be more specific. Abstract-I appreciate the limits on word count, but abstract may benefit from including group sample sizes, and also a sentence or two around possible explanations for the lack of between-group differences and implications/future directions. Introduction-Tables 1 and 2 are informative and very useful. It may be beneficial to also include year and country of publication for this high-level summary. Method-Page 28: PiCD does not seem to be introduced and defined before first use here. Please add this information. Results-Table 3: Typo “Personality Prait Scores”. Discussion-Page 40: Typo “that have detected groups differences”

Response: We appreciate the reviewer’s detailed review of the manuscript and suggested modifications. We agree that “Response Inhibition” is a more appropriate term of the title and have revised it as such. We have reworked the Abstract as suggested by including information on the objectives of the study, as well as broad overview of the methods and sample size information. We also included a brief explanation for the lack of between-group differences and implications for future direction (see p. 2). We have added years to Tables 1 and 2, but respectfully ultimately chose not include country due to ambiguity of the information presented in some of the original studies. The use of the PiCD acronym has now been corrected. Additionally, the identified typos have been corrected. 

Reviewer 2: Page 8: Could you be more specific regarding the use of term “Non-patients”: Do you refer to healthy controls only? This may be a better term, and is used in other places throughout manuscript (note a minor inconsistency as the term is written as “nonpatient controls” – without hyphen on p. 7.)

Response: The reviewer raises an important question that we agree requires clarification and consistency in language throughout the manuscript. We originally used the term “non-patients” to refer to study samples who did not explicitly look at a clinical sample (or who did not provide detailed explanations for their control groups) in the response inhibition literature review, and within our own study’s controls without BPD or MDD. However, this term increased ambiguity within the manuscript and may not adequately reflect the eligibility criteria for our study; therefore, we believe the more appropriate label would simply be “controls” (rather than “non-patients” or “healthy controls”), with the detailed information about the group’s eligibility criteria provided in the Methods section. Accordingly, we have changed the label for this group throughout the manuscript.

---

## [Decision Letter · Decision Letter 1]

23 Dec 2022

An Exploratory Study of Functional Brain Activation Underlying Response Inhibition in Major Depressive Disorder and Borderline Personality Disorder

PONE-D-22-21209R1

Dear Dr. Cane,

We’re pleased to inform you that your manuscript has been judged scientifically suitable for publication and will be formally accepted for publication once it meets all outstanding technical requirements.

Kind regards,

Jyrki Ahveninen

Academic Editor

PLOS ONE

Additional Editor Comments (optional):

Reviewers' comments:

Reviewer's Responses to Questions

**Comments to the Author**

1. If the authors have adequately addressed your comments raised in a previous round of review and you feel that this manuscript is now acceptable for publication, you may indicate that here to bypass the “Comments to the Author” section, enter your conflict of interest statement in the “Confidential to Editor” section, and submit your "Accept" recommendation.

Reviewer #1: All comments have been addressed

2. Is the manuscript technically sound, and do the data support the conclusions?

Reviewer #1: (No Response)

3. Has the statistical analysis been performed appropriately and rigorously? 

Reviewer #1: (No Response)

4. Have the authors made all data underlying the findings in their manuscript fully available?

Reviewer #1: (No Response)

5. Is the manuscript presented in an intelligible fashion and written in standard English?

Reviewer #1: (No Response)

6. Review Comments to the Author

Reviewer #1: I did already major and also minor suggestions. These all have been now answered carefully and thoroughly. Great and interesting work.

7. PLOS authors have the option to publish the peer review history of their article (what does this mean?). If published, this will include your full peer review and any attached files.

Reviewer #1: No

---

## [Editor Report · Acceptance letter]

28 Dec 2022

PONE-D-22-21209R1 

An Exploratory Study of Functional Brain Activation Underlying Response Inhibition in Major Depressive Disorder and Borderline Personality Disorder 

Dear Dr. Cane:

I'm pleased to inform you that your manuscript has been deemed suitable for publication in PLOS ONE. Congratulations! Your manuscript is now with our production department. 

Kind regards, 

on behalf of

Dr. Jyrki Ahveninen 

Academic Editor

PLOS ONE